# Variational Autoencoders with Normalizing Flow Decoders

## Abstract

Recently proposed normalizing flow models such as Glow (Kingma & Dhariwal, 2018) have been shown to be able to generate high quality, high dimensional images with relatively fast sampling speed. Due to their inherently restrictive architecture, however, it is necessary that they are excessively deep in order to train effectively. In this paper we propose to combine Glow model with an underlying variational autoencoder in order to counteract this issue. We demonstrate that such our proposed model is competitive with Glow in terms of image quality while requiring far less time for training. Additionally, our model achieves state-of-the-art FID score on CIFAR-10 for a likelihood-based model.

## 1 Introduction

The field of deep generative models is experiencing rapid progress, with new state-of-the-art results being achieved regularly. Currently, state-of-the-art in image generation is dominated by adversarial training methods such as Karras et al. (2019) and Brock et al. (2019). However, Generative Adversarial Networks (GANs) notoriously suffer from problems with training stability and mode collapse due to the dynamics of an adversarial objective. Many ideas have been proposed for dealing with these issues, such as Arjovsky et al. (2017) and Miyato et al. (2018), however issues inherently remain. Due to this, there is still signficant attention being given to pure likelihood models despite their poorer qualitative performance, as maximum likelihood grants stable training and is able to achieve good mode coverage on account of its heavy penalization of models that do not allocate probability mass to space containing training data.

There are numerous classes of likelihood-based deep generative models in the contemporary literature. Autoregressive models such as PixelRNN (van den Oord et al., 2016) have been shown to achieve impressive test likelihood scores on various datasets, however such models typically perform poorly when evaluated using quantitative measures of image quality, and in addition take a long time to generate samples. Variational autoencoders (VAE) (Kingma & Welling, 2014), a form of latent variable model and one of the earlier proposed deep generative models, are well known to produce samples of poor quality when implemented in their most basic form and as such require combining with more powerful generative models in order to be competitive. More recently, it was shown in Kingma & Dhariwal (2018) that a class of likelihood models known as *normalizing flows* can be used to generate high quality images with much faster sampling times than autoregressive models. Heavy restrictions must be imposed on the model architecture however in order to be able to efficiently calculate the determinant of the model Jacobian, forcing implementations of such models to be extremely deep and wide to make up for deficiencies of a restricted architecture. As a result, training requires an exorbitant amount of time; as an example 40 GPUs and ~1 week of training was required in order to produce the 256x256 CelebA-HQ samples in Kingma & Dhariwal (2018). Finding ways to cut down on these resource requirements is therefore pertinent in order to make such models accessible to the average practitioner. Towards achieving this goal, in this work we propose combining the standard variational autoencoder model with an overlying Glow layer and demonstrate state-of-the-art FID (Heusel et al., 2017) results while simultaneously achieving much lower training time.

## 2 BACKGROUND AND RELATED MODELS

### 2.1 VARIATIONAL AUTOENCODERS

Variational Autoencoders (VAEs), first proposed in Kingma & Welling (2014) are a popular class of deep latent variable model inspired by traditional variational inference. Given a dataset $X = \{x^{(1)}, x^{(2)}, ..., x^{(n)}\}$, the goal is to maximize the log-likelihood lower bound

$$\mathbb{E}_{q(z|x^{(i)})}[\log p(x^{(i)}|z)] - D_{KL}[q(z|x^{(i)})||p(z)] \tag{1}$$

where $q(z|x)$ is an encoder distribution and $p(x|z)$ is a decoder distribution, both parameterized by deep neural networks. The prior distribution $p(z)$ may be fixed or learned. Typically the encoder, decoder and prior distributions are all assumed to be some simple distribution, usually Gaussian. In this case the model enjoys the properties of stable training and fast convergence, as well as fast sampling. VAEs are well known to produce sample images of poor quality, however, owing to a number of previously identified factors.

It has been pointed out that the typical Gaussian assumption of the posterior $q(z|x)$ leads to poor posterior approximations, as the true posterior is often far from Gaussian in shape. As the optimizer is forced to fit mismatching distributions, this results in the marginal encoder distribution $q(z)$ diverging from the prior $p(z)$. In Jimenez Rezende & Mohamed (2015) and many subsequent works such as Kingma et al. (2016), it was proposed that the posterior be made more flexible via the use of normalizing flows, which are discussed in detail in the next section. In this way, the approximate posterior can achieve a better fit to the true posterior.

In Dai & Wipf (2019) the authors pointed out that the failure to optimize the decoder variance in typical implementations of VAEs is a major cause of poor quality samples, and that by optimizing decoder variance it is possible to achieve an optimal VAE cost in the case where the encoder and decoder distributions are Gaussian (specifically, a diagonal Gaussian encoder and isotropic Gaussian decoder). Achieving the optimal VAE cost does not necessarily recover the ground truth distribution, though, and so they additionally proposed learning the distribution of the latent space using a second-stage VAE, and proved that by doing so it is theoretically possible to recover the ground truth distribution.

Despite there being a theoretically obtainable optimal cost in the case where the decoder variance is learned, in practice using a decoder that assumes independence between pixels is not ideal, as it is well understood that this leads to blurry reconstructions due to the optimizer favouring solutions that are a weighted mean of data points in pixel space. Additionally, because Gaussian decoders are a poor fit for natural images, they may introduce substantial salt-and-pepper noise into the generated samples, and so it is commonplace to restrict decoder sampling to the mean. This artificially restricts the generative distribution to a manifold of dimensionality equal to that of the latent space, and potentially limits sample diversity. It is precisely these deficits of simple decoder distributions used in standard VAE models that we attempt to address in our proposal.

Complex, non-Gaussian decoder distributions have been proposed in several previous works. In Gulrajani et al. (2017), the authors propose using autoregressive PixelCNN layers (Van den Oord et al., 2016) in the decoder, and are able to achieve impressive test likelihood scores. An adversarially trained autoencoder was proposed in Larsen et al. (2016), with the decoder objective being defined by a standard GAN loss over the generated image plus a representation loss using L2 distance between discriminator hidden features. However, such a model cannot be used to evaluate likelihood, and being an adversarial method it suffers from problems with training stability. Shmelkov et al. (2019) propose a very similar model to our own, with a Real NVP (Dinh et al., 2016) normalizing flow model stacked on top of a VAE with Gaussian decoder. They combine likelihood maximization with adversarial training to achieve competitive FID scores on various datasets, however their model achieves poor results when using a purely likelihood based approach.

### 2.2 NORMALIZING FLOWS

Normalizing flows take advantage of the change of variables formula for probability distributions, given by

$$p(x) = p(z)|\det(\partial f(x)/\partial x^T)| \tag{2}$$

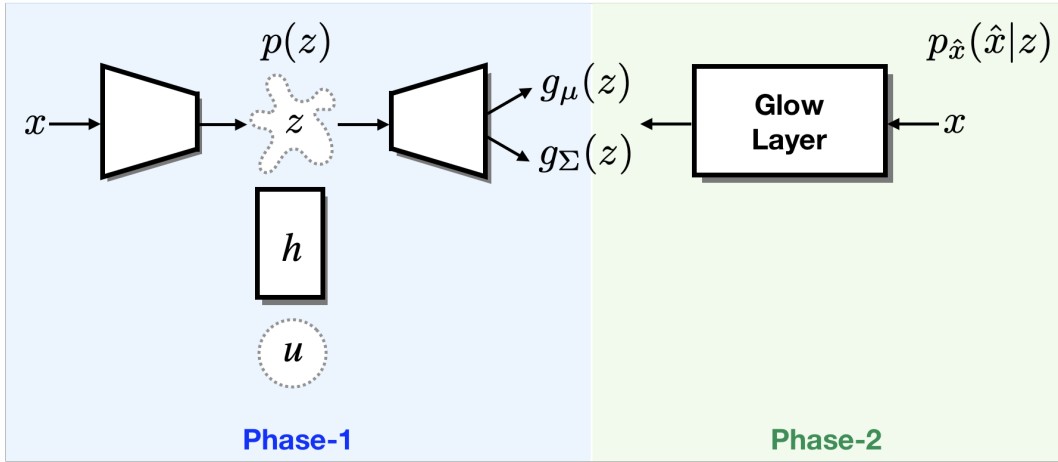

Figure 1: Overview of our proposed method, which combines a simple VAE with a normalizing flow model in the pixel space. The model training is split into two phases, please refer to Section 3.4 for more details.

where $f$ defines a bijection between the spaces of $x$ and $z$. They have found use with application to VAEs in order to achieve flexible priors/posteriors (Jimenez Rezende & Mohamed, 2015; Chen et al., 2017), as well as being used as standalone generative models directly in data space (Dinh et al., 2014) in which case exact likelihood evaluation is possible. One of the most common implementations is that of autoregressive flows, whereby the bijection $f$ is made to be auto-regressive resulting in a triangular Jacobian. This is important for efficiency, as it allows the determinant in Eq. 2 to be calculated in $O(n)$ rather than $O(n^3)$. Composition of multiple autoregressive functions allows for the construction of high capacity models.

## 3 METHOD AND MODEL DEFINITION

At a high level, our idea is to combine a simple VAE model with a normalizing flow model in the pixel space. As such, we hope to obtain the benefits of both: the smaller model size, faster training, and compact latent representation of VAEs, as well as the high quality samples of normalizing flows. We use a simple VAE to learn a base distribution, and train a normalizing flow using this base distribution, such that we end up with a conditional normalizing flow. As the VAE should be able to learn a distribution that is already close to the ground truth, our normalizing flow, when conditioned on this base distribution, should not need to do as much work as a full marginal normalizing flow model such as Glow (Kingma & Dhariwal, 2018). Our normalizing flow therefore need not be as deep as such models, saving training time and model size. Our model is trained using pure likelihood maximization, and is capable of evaluating a lower bound on the likelihood of arbitrary data samples.

For our approach we use an underlying Gaussian decoder to learn the base distribution of a normalizing flow. We denote the random variable produced by this underlying decoder as

$$\bar{x} \sim p_{\bar{x}}(\bar{x}|z) = \mathcal{N}(\bar{x}|g_\mu(z), g_\Sigma(z)) \tag{3}$$

where $g_\mu$ decodes the mean of $p_{\bar{x}}(\bar{x})$ and $g_\Sigma$ decodes the diagonal covariance.
The normalizing flow layer then defines a bijection $f : \hat{\mathcal{X}} \to \bar{\mathcal{X}}$ and random variable

$$\hat{x} \sim p_{\hat{x}}(\hat{x}|z) = p_{\bar{x}}(f(\hat{x})|z)|\det(\partial f(\hat{x})/\partial \hat{x}^T)| \tag{4}$$

where $\hat{x} \in \hat{\mathcal{X}}$ and $\bar{x} \in \bar{\mathcal{X}}$.

### 3.1 UNDERLYING GAUSSIAN DECODER

In Dai & Wipf (2019) the authors propose to add a learnable parameter $\gamma$ to the model, with the decoder likelihood given by $\mathcal{N}(x|g_\mu(z), \gamma \boldsymbol{I})$, and prove that doing so enables the model to always be

capable of achieving a better VAE cost by lowering the value of $\gamma$. This has an additional benefit in the case of our model, as it facilitates training of the overlying Glow layer. By allowing the VAE decoder to shrink its variance as it gains confidence in its estimates, the Glow layer has to do less work to remove any erroneous noise in the decoder distribution.

We extend this idea by allowing the decoder distribution to be Gaussian with diagonal covariance, rather than simply an isotropic Gaussian. By doing so, we allow the decoder to express confidence in its estimates at the pixel level. In the case of MNIST, for example, the decoder is likely to be highly confident in its estimate of the uniformly black background pixels, while the pixels on the border of the digit are likely to be the source of lower confidence. When sampling from the underlying VAE (while keeping decoder samples restricted to the mean) this does not result in any significant improvement over the simpler isotropic Gaussian decoder. For our full model, however, we found in our experiments that this small change resulted in significant improvements to FID score.

## 3.2 PRIOR DISTRIBUTION

As noted previously, we allow the decoder to learn the variance of its distribution. However, as shown in Dai & Wipf (2019), this results in the decoder likelihood quickly dominating the KL divergence term, resulting in the model favouring learning of the data manifold over learning the ground truth distribution. In other words, the model is able to learn much more accurate reconstructions at the expense of the encoder distribution matching the prior. Their proposed solution is to learn the encoder distribution using a second-stage VAE, and they prove that by doing so they are theoretically capable of recovering the ground truth distribution. We note, however, that by using a flexible learned prior in the form of an autoregressive normalizing flow, we are able to calculate exact prior likelihood and are also able to recover the ground truth distribution in a single stage of training. Recovery of the ground truth distribution is possible as autoregressive normalizing flows are universal approximators (Huang et al., 2017). This gives us a second latent distribution $p(u) = \mathcal{N}(u|0, \boldsymbol{I})$ and bijection $h : \mathcal{Z} \rightarrow \mathcal{U}$ with $z \in \mathcal{Z}$ and $u \in \mathcal{U}$. The prior distribution is then given by $p(z) = p(u)|\det(\partial h(z)/\partial z^T)|$.

Such a model was originally proposed in Chen et al. (2017), where the authors show that it is equivalent to Inverse Autoregressive Flow (IAF) (Kingma et al., 2016) along the encoder path, while being deeper along the decoder path. We therefore obtain the benefit of using a flexible posterior while simultaneously being able to close the gap between the encoder and prior distributions. We found in our experiments that by using a normalizing flow prior we were able to obtain better FID scores than by using a second-stage VAE. Results are shown in the experiment section.

In our implementation we use a normalizing flow similar in structure to Real NVP Dinh et al. (2016) (which is a special case of autoregressive normalizing flows Papamakarios et al. (2017)), as it allows both efficient training and sampling.

## 3.3 NORMALIZING FLOW LAYER

We use the additive version of the Glow model proposed in Kingma & Dhariwal (2018). More specifically, our flow is comprised of several blocks, with each block containing an activation normalization followed by an invertible 1x1 convolution, which is then followed by an additive coupling layer (Dinh et al., 2014). Since our flow does not need to be as deep on account of it being conditioned on the underlying VAE distribution, we do not utilize any split or squeeze operations.

## 3.4 TRAINING

We split the model training into two phases. In the first phase, we train the underlying VAE and its prior. In the second phase, we keep all components of the underlying VAE including its prior fixed while training the Glow layer on top of it. That is, in the second stage we optimize the parameters of the Glow layer to maximize $\mathbb{E}_{q(z|x^{(i)})}[\log p_{\hat{x}}(x^{(i)}|z)]$. This continues to maximize likelihood, as the first term in Eq. 1 is maximized while the second term is kept fixed. It is also possible to train the entire model jointly in one phase, however we found that this resulted in poorer quality images, likely because the Glow layer is unable to train efficiently with a changing base distribution.

## 3.5 Sampling

Sampling from the model is performed straightforwardly by first sampling $\bar{x}$ from the underlying marginal VAE distribution, and then calculating the final sample via $f^{-1}(\bar{x})$. In Kingma & Dhariwal (2018), the authors make use of reduced-temperature sampling in order to improve image quality. We note however that, since this operation results in significantly reduced sample diversity, it will have a negative impact on distance measures such as the FID score. In our model, because the underlying VAE is trained first and then subsequently kept fixed during training of the Glow layer, the VAE ends up being responsible for sample diversity while the Glow layer "sharpens" the VAE output. Due to this, we are able to perform temperature reduction in the Glow layer without significantly sacrificing sample diversity. We found that using a temperature of $0.5$ in the Glow layer produced the highest FID scores across all of our experiments.

## 4 Experiments

### 4.1 Normalizing flow prior vs two-stage VAE

We begin our experiments by first demonstrating our choice of normalizing flow prior is advantageous over the second-stage VAE proposed in Dai & Wipf (2019). We conduct experiments on Fashion-MNIST (Xiao et al., 2017) and CIFAR-10 (Krizhevsky et al., 2009) using decoder distribution given by $\mathcal{N}(x|g_\mu(z), \gamma \boldsymbol{I})$, where $\gamma$ is a learnable parameter. The experimental results are shown in Table 2 where the benefit of using normalizing flow prior in our proposed method is well provided, in comparison to the 2-stage VAE (Dai & Wipf, 2019). Please note that here we especially use the same network architecture for (Dai & Wipf, 2019) and the VAE of our proposed model, in order to have fair comparison.

|  | FID |
| --- | --- |
| Fashion (2nd VAE) | 31.87 |
| Fashion (Flow) | 19.70 |
| CIFAR-10 (2nd VAE) | 76.04 |
| CIFAR-10 (Flow) | 71.85 |

Table 1: FID scores comparing a normalizing flow prior with a second-stage VAE.

### 4.2 Results for full model

We now present our experimental results on both the CIFAR-10 (Krizhevsky et al., 2009) and CelebA (Liu et al., 2015) datasets. For both datasets we use a single layer Glow (i.e. no squeeze or split operations) of depth 32. For the underlying VAE we use the Resnet model described in Dai & Wipf (2019), with a depth of 3 for CIFAR-10 and a depth of 4 for CelebA. The prior was implemented using a 16 layer normalizing flow with each layer comprised of two fully connected layers of width 1024, each followed by a ReLU. For both datasets we trained the underlying VAE for 1,000 epochs before training the Glow layer also for 1,000 epochs. We use an initial learning rate of 0.0001 and halve it every 250 epochs. In the case of CIFAR-10, we found that retraining the prior with early stopping after training the VAE helped to improve FID. Early stopping was done after 50 epochs. We note that we do not make use of data augmentation in any of our experiments.

### 4.2.1 Quantitative evaluation

We report Fréchet Inception Distance (Heusel et al., 2017) calculated against a held-out test set for quantitative image quality evaluation, as well as test bits per dimension. For our CIFAR-10 results we include FID scores and bits-per-dimension reported in previous works involving likelihood-based models, in order to provide a more comprehensive list of baselines; results were taken from Chen et al. (2019) and Ostrovski et al. (2018). In Chen et al. (2019) the authors report an FID score of 46.90 for Glow, however after training the official Glow model ourselves using additive coupling layers for 2,000 epochs we achieved an FID score of 45.23, so we report this in our table instead. Results for CIFAR-10 can be found in Table 2, and results for CelebA can be found in Table 3. We achieve

|            | FID   | bits/dim   |
|------------|-------|------------|
| PixelCNN   | 65.93 | 3.03       |
| PixelIQN[†]| 49.46 | -          |
| Glow       | 45.23 | 3.35       |
| Residual Flow | 46.37 | 3.28   |
| Ours       | 42.14 | $\leq 3.17$ |

Table 2: FID score and bits per dimension on CIFAR-10.
(† uses quantile regression)

|      | FID   | bits/dim |
|------|-------|----------|
| Glow | 19.00 | 2.25     |
| Ours | 20.64 | 2.81     |

Table 3: FID score on CelebA

what is to our knowledge state-of-the-art FID on CIFAR-10 for a pure likelihood model, while also outperforming Glow on test likelihood. For CelebA dataset, we have competitive FID to the Glow model, but with much less time for training, as what is going to be discussed in the Section 4.3.

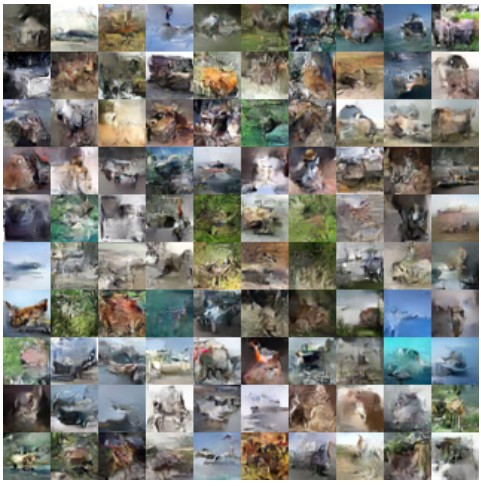

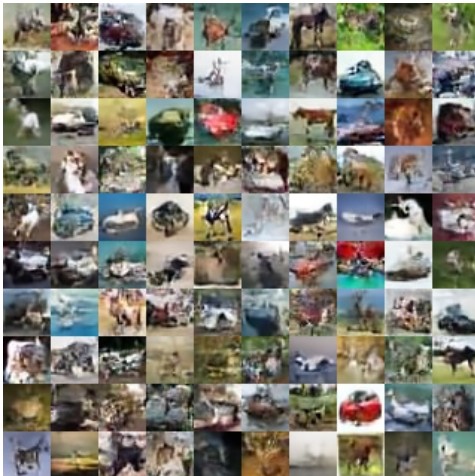

(a) Glow (Kingma & Dhariwal, 2018)  (b) Ours

Figure 2: Samples from models trained on CIFAR-10.

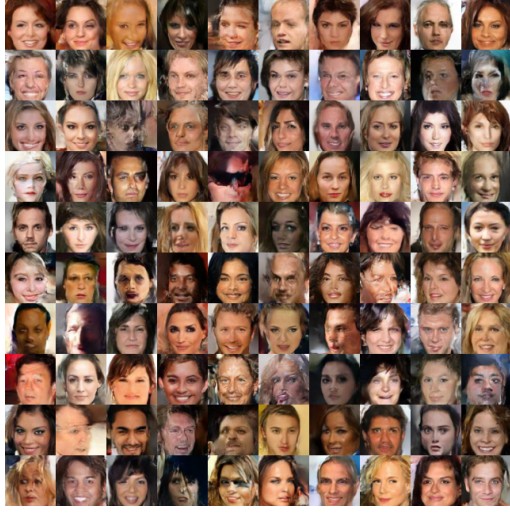

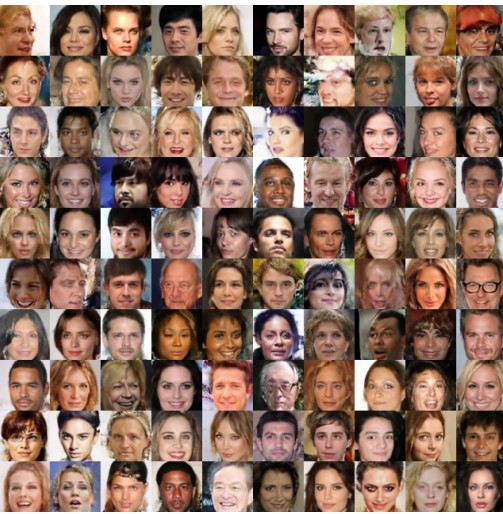

(a) Glow (Kingma & Dhariwal, 2018)  (b) Ours

Figure 3: Samples from models trained on CelebA.

### 4.2.2 QUALITATIVE EXPERIMENTS

First, some qualitative results for CIFAR-10 and CelebA datasets are provided in Figure 2 and Figure 3 respectively, based on the Glow model and our proposed one. We can see that our proposed model is able to produce competitive or better results in terms of image quality and diversity. Then, we present qualitative experiments to demonstrate the behaviour of our model. As mentioned earlier, the Glow layer essentially becomes a "sharpening" layer as a result of our two-stage training procedure. To demonstrate this, in Figure 4 we show random samples from the VAE alongside samples from the same latent code after having been passed through the Glow layer. It can be seen that the VAE samples are somewhat blurry as expected, and are considerably sharper after being passed through Glow. The content is entirely preserved by the Glow layer, with stochasticity resulting in small changes to the fine details that are only visible upon close inspection.

Additionally, we show images obtained by interpolating between randomly sampled points in the VAE latent space in Figure 5. This is common practice for latent variable models in the literature to demonstrate the model's ability to generalize and learn a useful compact representation.

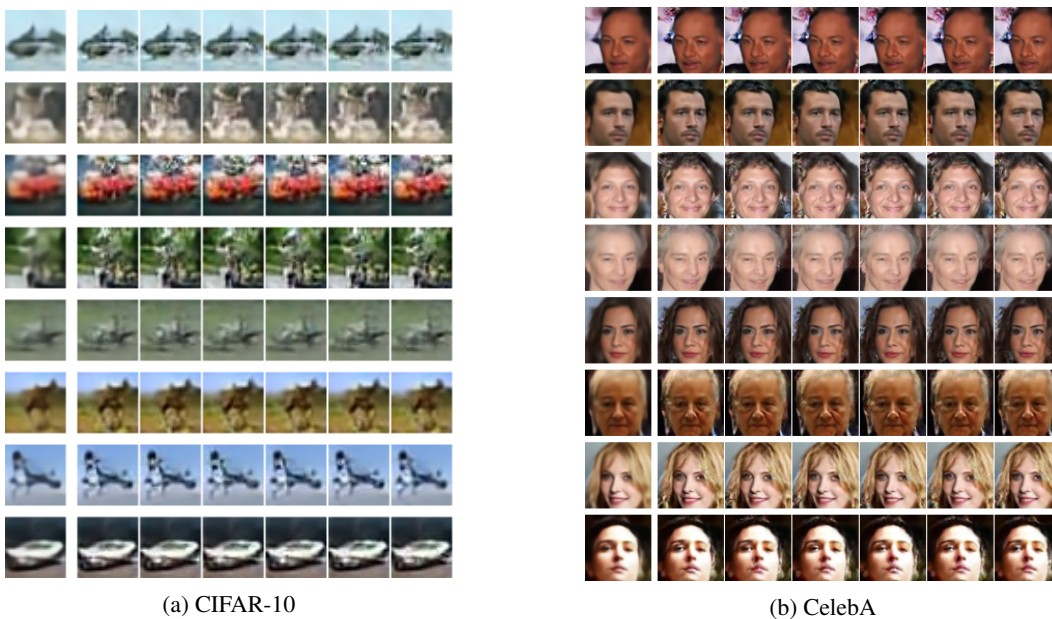

(a) CIFAR-10

(b) CelebA

Figure 4: Example of how the Glow layer modifies VAE samples. Left column: samples from the underlying VAE. Right columns: decoder samples after being passed through the Glow layer.

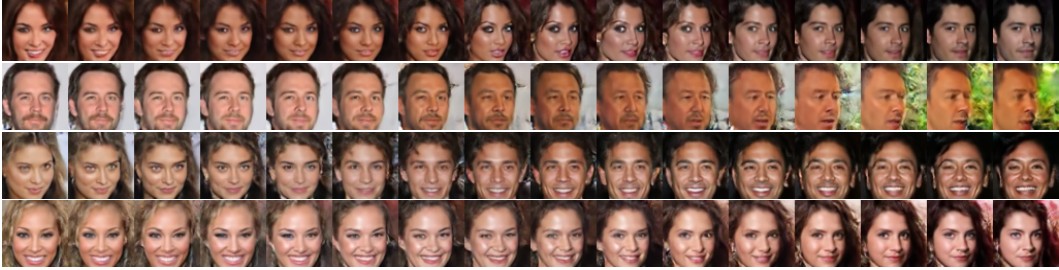

Figure 5: Interpolation between randomly sampled points in the VAE latent space, demonstrating that our model learns a useful compact representation.

### 4.3 TRAINING TIME

One of the main claims made regarding our model is that it can significantly reduce training time when compared with a standalone Glow model. In order to validate this claim, we measure wall clock

| CIFAR-10 | |
|---|---|
| | Time/epoch (seconds) |
| Glow | $\sim 512$ |
| Ours (VAE) | $\sim 118$ |
| Ours (Glow) | $\sim 162$ |
| Ours (Avg.) | $\sim 140$ |

Table 4: Training time per epoch for different models (dataset:CIFAR-10).

time per epoch when training each model and report the results in Table 4. Times were measured using a GeForce GTX 1080 Ti. We measure VAE and Glow layer times separately for our model and, since we train an equal number of epochs for each, include the average of the two for the overall time per epoch. We make a note of the fact that we did not use gradient checkpointing when evaluating training time. Results indicate that our model trains over 3.5x faster than Glow.

## 5 CONCLUSION AND FUTURE RESEARCH

We have presented a model combining standard VAEs with Glow, and demonstrated that it is capable of achieving better sample image quality than a standalone Glow model while being significantly faster to train. It is our hope that our results are useful to any future works attempting to bridge the gap between image quality of likelihood-based models and GANs. Future research directions may involve investigating whether our results continue to hold when scaling up to higher dimensional images.

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
