# OpenReview forum: "Variational Autoencoders with Normalizing Flow Decoders"
_ICLR.cc/2020/Conference — Reject_

### Official Review · AnonReviewer1 · 2019-10-22
**Official Blind Review #1**

**Rating:** 3

**Review:**

The paper proposes a combination of a conditioned flow-based model with a VAE. The main contribution of the paper is a two-phase training that allows to train the model. Unfortunately, a joint training of the model failed. In general, combining VAEs with flow-based models is an important research direction. Unfortunately, the paper is not clearly written. A lot of details are missing that makes the paper impossible to reproduce and fully understand. Moreover, the most interesting part about training procedure, is discussed superficially. Finally, I find the comparison to SOTA methods disappointing. The authors skipped many recent papers. I do not expect to see SOTA results among all models, but at least comparable results within a group of approaches. However, the authors provide only a small subset of models that makes me wonder whether they are aware of other papers.

Remarks:
- The following statement is not fully correct:
"In our implementation we use a normalizing flow similar in structure to Real NVP Dinh et al. (2016) (which is a special case of autoregressive normalizing flows Papamakarios et al. (2017)), as it allows both efficient training and sampling"
Real NVP is a bipartite flow, while Masked Autoregressive Flow is an autoregressive flow. In special cases, these two flows coincide, but their original implementations are different.

- The paper misses a lot of important details. For instance, a reader needs to figure out what is the objective function. Further, the authors do not mention how they deal with images represented by integers. Do they use dequantization as in other papers (e.g., Theis, L., van den Oord, A., and Bethge, M.  A note on the evaluation of generative models. In Workshop Track,ICLR, 2016)?  These are very important details to fully understand the proposed approach. Providing a very general diagram (Figure 1) and generic equations (2-4) are not sufficient. Currently, there are different implementations of flow components, and describing them would be definitely beneficial. Moreover, it is important to show how conditioning is used in the flow model.

- Section 3.4 is incredibly interesting part of the paper and it should be further analyzed. The proposed two-phase is reasonable, but it leaves an open question why a joint training fails. I agree with the authors that a possible explanation is training instability. Nevertheless, I would be more than interested in seeing a more thorough analysis of this phenomenon.

- I do not understand why the authors skipped reporting bpd for CelebA.

- In general, the comparison in Table 2 is far from being complete. For instance, please see the following paper:
Ho, J., Chen, X., Srinivas, A., Duan, Y., & Abbeel, P. (2019). Flow++: Improving flow-based generative models with variational dequantization and architecture design. arXiv preprint arXiv:1902.00275.
On Cifar10, current non-autoregressive models are able to achieve 3.08 bpd (Flow++)  and 3.11 bpd (IAF-VAE). This is much better than the reported 3.17 bpd.

- In Section 4.2.2, first line, a number to a figure is missing.

======== AFTER REBUTTAL ========
I would like to thank the authors for their rebuttal. However, I am not fully satisfied with some answers. First of all, the paper should be clearly written so that a reader could be able to find all necessary details and seamlessly implement the presented idea. If some details do not fit the main text, then they should be included in an appendix. Second, the objective function is an important component of any ML problem and, therefore, it should be included in the paper. Sometimes a verbal description is not sufficient. Third, by conditioning a flow I meant whether the based distribution was conditional or/and a single flow layer was conditioned on z. Currently, there are multiple ways to condition and I was curious which of them was used by the authors. Fourth, I do not believe that achieving nicely looking images is the ultimate task for generative models. However, this is an open discussion. Fifth, I agree, the discussion on why the joint training failed is an incredibly interesting question.
Again, I really appreciate all the answers, and I believe that the authors did their best to improve the paper. However, as a reviewer I must look for novelty and evaluate how the paper is readable for others. In my opinion, combining VAE with GLOW is not sufficiently novel idea. If the paper was very clearly written, I would think about rising the paper. However, right now, I decide to keep my original score.

**Experience Assessment:**

I have published in this field for several years.

**Review Assessment: Checking Correctness Of Derivations And Theory:**

I assessed the sensibility of the derivations and theory.

**Review Assessment: Checking Correctness Of Experiments:**

I assessed the sensibility of the experiments.

**Review Assessment: Thoroughness In Paper Reading:**

I read the paper at least twice and used my best judgement in assessing the paper.

---

> ### Author Response · Authors · 2019-11-15
> **Response to Reviewer 1**
>
> We thank the reviewer for their valuable feedback..
>
>
> > In special cases, these two flows coincide, but their original implementations are different
>
> We are not quite sure how this statement differs from ours. Also, quoting directly from the MAF paper "the coupling layer used in Real NVP is a special case of the autoregressive layer used in MAF."
> In any case, the remark isn't very important to our paper.
>
>
> > For instance, a reader needs to figure out what is the objective function
>
> We believe the objective function is already communicated. In words, it is a VAE objective (Eq. 1) with the decoder distribution given by a Glow model (section 3.3) with base distribution given by a diagonal Gaussian conditioned on the latent code. (Eq. 3 and 4). The prior distribution is given by a normalizing flow (section 3.2). Perhaps it could be communicated more clearly, but the information is there.
>
>
> > Further, the authors do not mention how they deal with images represented by integers
>
> We use the same dequantization method as that used by Glow, i.e. for discrete pixel values in the range [0, 256) we add uniform noise in the range [0, 1).
>
>
> > Currently, there are different implementations of flow components, and describing them would be definitely beneficial
>
> We use exactly the same implementation as Glow for the decoder flow. We acknowledge that information about the flow used in the prior distribution could be more thorough.
>
>
> > Moreover, it is important to show how conditioning is used in the flow model.
>
> Does this refer to conditioning on the VAE latent code, or conditioning on the other dimensions within the flow?
> If it's the former, this should be clear from Equation 4. If it's the latter, we use exactly the same conditioning as Glow.
>
>
> > Section 3.4 is incredibly interesting part of the paper and it should be further analyzed
>
> We agree that our discussion of this phenomenon is superficial and that a more thorough analysis is warranted.
> Unfortunately we have been unable to produce any satisfactory results before the discussion deadline and so we will endeavour to give more attention to this section in future revisions.
>
>
> > On Cifar10, current non-autoregressive models are able to achieve 3.08 bpd (Flow++)  and 3.11 bpd (IAF-VAE)
>
> Our main concern is image quality, with which likelihood is not totally correlated. For example PixelCNN achieves
> better BPD than both models you mentioned, but its generated images are not high quality due to over-generalization. Therefore we prefer to focus on FID score. Please see our response to Reviewer 3 regarding Flow++.

---

### Official Review · AnonReviewer2 · 2019-10-23
**Official Blind Review #2**

**Rating:** 6

**Review:**

This paper proposed a variant of the variational autoencoder, specifically by using a simplified normalizing flow model as the decoder. When compared with Glow, the proposed method is simpler and more efficient to train. The authors applied their algorithm on a few datasets, and showed better or competitive performance, both qualitatively and quantitatively.

This paper is in general well written. I think the idea looks interesting, although the novelty is a bit incremental, as it basically combined the two well-known models (VAE and Glow). The experimental results showed the promise of the new method, which could be more convincing if applied on larger scale datasets. My detailed comments and questions are as follows.
1. Regarding training, the authors decomposed it into two phases, i.e., training VAE first and then Glow. The authors also mentioned that jointly training resulted in images with poor qualities. I am curious about how the authors designed the Glow model: Intuitively a larger model may have more modeling capacity, but at the cost of computational cost. Some ablation studies or explanation could be helpful.
2. The authors claimed at the beginning of Section 3 that the their normalizing flow "should not need to do as much work as a full marginal normalizing flow model such as Glow". I am wondering how the performance will be for the Glow used, if without the VAE part?
3. For the bits/dim results for Glow in Table 2, was it computed by yourself or just from the Glow paper? I saw the FID score was obtained by yourself.
4. For the Glow used in experiments, how does its architecture compare with the one used in the original paper?
5. I am a bit surprised to see the results in Table 3 and Figure 3, as Glow has a better FID score but the overall image quality is worse. Is it related to the size of the Glow used?

**Experience Assessment:**

I have published one or two papers in this area.

**Review Assessment: Checking Correctness Of Derivations And Theory:**

N/A

**Review Assessment: Checking Correctness Of Experiments:**

I carefully checked the experiments.

**Review Assessment: Thoroughness In Paper Reading:**

I read the paper thoroughly.

---

> ### Author Response · Authors · 2019-11-15
> **Response to Reviewer 2**
>
> We thank for the reviewer for their valuable feedback.
>
>
> > I am wondering how the performance will be for the Glow used, if without the VAE part?
>
> Without the VAE part it will perform very poorly, since it would be only the final layer of the original Glow model which has only 12 channels and so would not be able to effectively capture long range dependencies or model high level structure.
>
>
> > For the bits/dim results for Glow in Table 2, was it computed by yourself or just from the Glow paper?
>
> The bits/dim was taken from the Glow paper. We only calculated FID ourselves since it is not reported in the original paper.
>
>
> > For the Glow used in experiments, how does its architecture compare with the one used in the original paper?
>
> It is exactly the same except for being only a single layer on top of the VAE, whereas in the original Glow they use 3 layers for CIFAR-10 with squeeze/split operations inbetween each.
>
>
> > I am a bit surprised to see the results in Table 3 and Figure 3, as Glow has a better FID score but the overall image quality is worse. Is it related to the size of the Glow used?
>
> Apologies, we need to do more investigation to understand why our proposal performs worse on FID for the CelebA dataset.

---

### Official Review · AnonReviewer3 · 2019-10-27
**Official Blind Review #3**

**Rating:** 3

**Review:**

This paper proposes adding additional flow layers on the decoder of VAEs. The authors make two claims
1. The proposed model achieves better image quality than a standalone Glow.
2. The proposed model is faster to train than Glows.
The intuition is a VAE can learn a distribution close enough to be target distribution, and the Glow only needs to do much less work than standalone Glow, hence faster. Some positive results are reported in the experiments, including better image quality, faster training time, and the Glow indeed sharpens the output of VAEs.

The paper indeed has some good results, particularly they can achieve it only with single-scale Glows with additive coupling layers. However, I think the claims are not sufficiently supported. Taking point 1 as an example, it is not clear to me why VAE+Glow is better than a standalone Glow. Imagine two models

M1: VAE+Glow (proposed in the paper)
M2: Glow0+Glow (standalone Glow)

sharing the last "Glow" part. M1 is better than M2 implies "VAE" is more powerful than "Glow0", which I doubt. Similarly, for point 2, it is not clear to me why "VAE" is faster than "Glow0". I think comparing the proposed model with IAF make more sense, because the proposed model just adds flows to the decoder and the prior. But the relationship with Glows needs to be considered more thoroughly.

Another confusing detail for me is the two-stage training in Sec. 3.4. The explanation "likely because the Glow layer is unable to train efficiently with a changing base distribution" doesn't make sense. Because IAF does successfully train their q-net without 2-stage training. There might be other reasons?

The baselines are not strong enough. Most importantly, Flow++ [1] reports a likelihood 3.08 on Cifar10 with standalone flows, which should also be a part of the baseline. I also wonders whether the proposed model benefits from deeper model, like standalone flows do. Will standalone flows surpasses the proposed model as the number of layers goes to infinity?

[1] Ho, Jonathan, et al. "Flow++: Improving flow-based generative models with variational dequantization and architecture design." arXiv preprint arXiv:1902.00275 (2019).

Finally, the paper is somewhat incremental. Particularly comparing with VAE-IAF, where this paper just adds flow layers to not only q but also p.

Update
=====

After reading the rebuttal I found some of my concerns are unaddressed. (regarding to the two-stage training, and the novelty)

Point 1 is still not answered. The answer I expect to have is how "The interaction between two models when they are being stacked may affect the overall performance in such a way that it is more than just the sum of its parts." Why are these two models perform particularly well when combined? The purpose of my initial question is for some in-depth analysis and intuition / theory.

Therefore I will keep my score unchanged.

**Experience Assessment:**

I have read many papers in this area.

**Review Assessment: Checking Correctness Of Derivations And Theory:**

I carefully checked the derivations and theory.

**Review Assessment: Checking Correctness Of Experiments:**

I assessed the sensibility of the experiments.

**Review Assessment: Thoroughness In Paper Reading:**

I read the paper thoroughly.

---

> ### Author Response · Authors · 2019-11-15
> **Response to Reviewer 3**
>
> We thank the reviewer for their valuable feedback.
>
>
> > sharing the last "Glow" part. M1 is better than M2 implies "VAE" is more powerful than "Glow0"
>
> If "Glow0" is all layers except the last in the original Glow implementation (which is the comparison being made in our paper), then this is indeed true since "Glow0" would only model half the number of dimensions of the image with the other half being purely Gaussian noise.
>
> If you are instead suggesting to add another layer to the original Glow implementation (effectively doubling the depth of the final layer), then we think it would be a more fair comparison to also double the depth of the Glow layer on top of the VAE. At this point however such excessively large models become impractical for common use.
>
> Furthermore, we don't agree with the statement that M1 is better than M2 implies "VAE" is more powerful than "Glow0". The interaction between two models when they are being stacked may affect the overall performance in such a way that it is more than just the sum of its parts.
>
>
> > it is not clear to me why "VAE" is faster than "Glow0"
>
> The reason for this is that the architecture of VAEs is not constrained in any way, unlike that of Glow which is constrained to have triangular Jacobian. Because of this, VAEs are not required to be nearly as deep to achieve the same expressiveness.
>
>
> > I think comparing the proposed model with IAF make more sense
>
> The underlying VAE of our model is for the most part equivalent to VLAE [1], which is equivalent to a deeper form of IAF. Therefore comparing our proposed model to IAF would be similar to comparing our full model to just the underlying VAE component of our model, which we have already done in Figure 4. The decoder likelihood of IAF assumes independence between pixels, and so results are not as sharp.
>
>
> > Flow++ [1] reports a likelihood 3.08 on Cifar10 with standalone flows, which should also be a part of the baseline
>
> We would be happy to include Flow++ in our baselines. The main focus of our paper however is image quality and hence we consider FID score more important, and we cannot find a reported FID score for Flow++ and so we would have to train their model ourselves which we did not have time/resources to do before the submission deadline.
> Also, if we were to compare against Flow++ we think it would make sense to stack a Flow++ layer on top of a VAE for comparison. We will certainly consider this approach in any future revisions.

---

### Public Comment · ~Zhisheng_Xiao1 · 2019-11-02
**Connection to an existing literature**

Hi,

I wonder if you have seen Deep Variational Inference Without Pixel-Wise Reconstruction (https://arxiv.org/pdf/1611.05209.pdf). I think the picture that illustrates the model in these two works are very similar. Could you please explain their difference?

Thanks!

---

> ### Author Response · Authors · 2019-11-15
> **Response**
>
> Thank you for your comment. We were actually not aware of this paper before submission. Their model is indeed very similar. The main differences are the type of normalizing flow that is used in the decoder (we use Glow, they use Real NVP), the training method (we use a two-stage approach) and the prior distribution which is learned in ours.

---

### Decision · Program_Chairs · 2019-12-19

**Decision:**

Reject

**Comment:**

The paper received mixed reviews: WR (R1,R3) and WA (R2). AC has carefully read reviews and rebuttal and examined the paper. Unfortunately, the AC sides with R1 & R3, who are more experienced in this field than R2, and feels that paper does not quite meet the acceptance threshold. The authors should incorporate the comments of the reviewers and resubmit to another venue.